# Self-management of type 2 diabetes mellitus in pregnancy and breastfeeding experiences among women in Thailand: Study protocol

**Ratchanok Phonyiam**[1,2]*, **Marianne Baernholdt**[3], **Eric A. Hodges**[1]

**1** School of Nursing, University of North Carolina at Chapel Hill, Chapel Hill, North Carolina, United States of America, **2** Ramathibodi School of Nursing, Faculty of Medicine Ramathibodi Hospital, Mahidol University, Ratchathewi, Bangkok, Thailand, **3** School of Nursing, University of Virginia, Charlottesville, Virginia, United States of America

* ratch@ad.unc.edu

## Abstract

Women with type 2 diabetes mellitus are at a higher risk of pregnancy complications. Although traditional beliefs and practices influence diabetes management and breastfeeding, recommendations integrating Thai cultural beliefs in maternal care are lacking. The purpose of this study is to describe diabetes self-management in pregnancy and breastfeeding experiences in women with preexisting type 2 diabetes mellitus from Thailand. A convergent parallel mixed-methods study will be conducted. Data will be collected from 20 pregnant women with preexisting type 2 diabetes mellitus in Thailand who are either primigravida or multigravida, aged 20–44 years old, speak the Thai language, and provide consent. The National Institute on Minority Health and Health Disparities Framework's sociocultural and behavioral domains guides the research aims. Data will be collected two times. The first time is during pregnancy (T1); study participants will complete questionnaires and engage in an interview about diabetes self-management, breastfeeding confidence, and breastfeeding intention. The second time is at 4–6 weeks postpartum (T2); study participants will be interviewed about their breastfeeding experiences. We will review and extract maternal health outcomes including body mass index, gestational weight gain, and glycated hemoglobin for T1 as well as fasting plasma glucose for T2. Qualitative data will be analyzed using directed content analysis. Quantitative data will be analyzed using descriptive statistics. Data sources will be triangulated with relative convergence in the results. This proposed study is significant because the findings will be used as a preliminary guide to developing a culturally tailored approach to enhance health outcomes of Thai women with diabetes in pregnancy and postpartum periods.

## Introduction

Globally, the diabetes prevalence was 10.5% (537 million adults) in 2021 [1]. Thailand is among the countries with a higher prevalence rate in women than men [2]. According to a recent Thai national survey, the diabetes prevalence rate rose from 8.3% in 2004 to 10.8% in

**Data Availability Statement:** Data sharing is not applicable to this article as no datasets were generated or analyzed. Data will be available after the completion of the study.

**Funding:** This study is funded by the Alpha Alpha Chapter of Sigma Theta Tau International Research Grant and the Sigma Small Grant. The funding body was not involved in the design of the study and collection, analysis, and interpretation of data or in writing this manuscript.

**Competing interests:** The authors declare that they have no competing interests.

2014 among women aged more than 20 years [2]. The number of women with type 2 diabetes mellitus (T2DM) who become pregnant has increased [1]. Pregnancy with T2DM is associated with a two-fold increased risk of pre-eclampsia [3] and a three-fold increased risk of both miscarriage and perinatal mortality [4]. Exposure to maternal complications contributes to higher risks of fetal macrosomia (birth weight of more than 4,000 grams), respiratory distress syndrome, and neonatal death [5]. These adverse outcomes can be minimized with proper diabetes self-management during pregnancy [6].

Thai traditional beliefs and practices influence diabetes management behaviors, serving as either barriers or facilitators [7, 8]. For example, tradition dictates that pregnant women must take many precautions, including avoiding certain foods and rigorous activity to ensure their unborn child's safety and well-being [9]. Further, typical Thai meals are served together with rice, so the restriction of carbohydrates in T2DM creates a unique challenge. Balancing dietary and physical activity to achieve optimal glycemic control is a challenge [6]. Research is needed to further explore how the Thai sociocultural environment influences women in managing diabetes during their pregnancy [8].

In addition to challenges for maternal health during pregnancy for women with T2DM, mothers with T2DM commonly experienced difficulty with milk supply, delayed lactation [10], and have worries about hypoglycemia episodes while breastfeeding [11]. Mothers with T2DM are less likely to establish exclusive breastfeeding at hospital discharge compared to women without T2DM [12]. Evidence showed that infants born to mothers with diabetes experienced complications, such as hypoglycemia and respiratory disorders, which often required neonatal intensive care unit admission [13]; this mother-infant separation decreases the chances of establishing breastfeeding. One study reported that infants born to mothers with diabetes were not interested in breastfeeding and had trouble sucking in the first two weeks compared to those born to mothers without diabetes [14].

Breastfeeding benefits have been widely recognized for mothers with diabetes. Early breastfeeding initiation optimizes maternal glucose regulation. A systematic review found that mothers who breastfeed their infants have either reduced or comparable insulin requirements than those who bottle-feed [15]. The mechanisms are threefold. First, suckling directly influences maternal blood glucose levels through glucose uptake into the breast or indirectly via lactogenic hormones such as oxytocin and prolactin [15]. Second, at each breastfeeding episode, oxytocin expresses into skeletal muscle via the oxytocin receptor and stimulates insulin secretion [16]. Finally, breastfeeding women had significantly higher prolactin levels than women who did not ($p < 0.001$) as prolactin acts directly on pancreatic beta-cells and increases insulin concentration [15]. Prolactin has been negatively correlated with HbA1c and 2-hour serum C-peptide ($r = -0.564$, $p = 0.005$, $r = -0.539$, $p = 0.008$, respectively) [17]. In addition to glucose regulation, breastfeeding mothers had lower triglycerides, higher HDL cholesterol [18], lower body mass index, and less weight circumference [19] when compared to non-breastfeeding mothers.

The World Health Assembly has endorsed a global target for increasing the exclusive breastfeeding (EBF) rate in the first six months up to at least 70% by 2030 [20]. In a 2019 survey, only 14% of all Thai infants aged 0–6 months were exclusively breastfed [21]. The six-month EBF rate in Thailand is still far from optimal [21], even though a Baby-Friendly Hospital Initiative was scaled up and implemented nationwide in 1992 [22]. To our knowledge, no studies have yet explored how having T2DM in pregnancy may affect Thai women's breastfeeding plans. Gaining a better understanding of Thai women's diabetes management experiences during pregnancy and breastfeeding practices postpartum will inform evidence-based clinical approaches to enhance the health of women with T2DM and their children in Thailand.

### Aims

This study is guided by the social and cultural domains in the National Institute on Minority Health and Health Disparities (NIMHD) Framework [23]. This framework provides an understanding of sociocultural and behavioral domains of influence at individual, interpersonal, and community levels. The specific aims are 1) to describe attitudes and confidence toward diabetes self-management in Thai pregnant women with preexisting T2DM, 2) to describe the barriers and facilitators of diabetes self-management in pregnancy, 3) to describe breastfeeding confidence and intention in pregnancy, and 4) to describe breastfeeding barriers and facilitators in women 4–6 weeks postpartum.

## Materials and methods

### Study design

This study uses a convergent parallel mixed-methods design [24]. Qualitative and quantitative data will be collected concurrently [24]. The rationale for this mixed methods design is to bring together the results of qualitative and quantitative data to obtain a more comprehensive understanding of women's diabetes self-management in pregnancy and breastfeeding experiences [24]. In this study, qualitative elements are the primary methods used to answer the research questions; at the same time, supplementary data from quantitative elements will deepen the findings [24]. The qualitative data will be derived from a qualitative descriptive approach [25, 26]. Qualitative description is an appropriate qualitative approach to answer our research questions because we will discover diabetes management and breastfeeding experiences from women with T2DM. The quantitative data will employ a cross-sectional descriptive design using self-reported questionnaires and physiological measures [27].

### Participant eligibility and recruitment

This study will recruit pregnant women with T2DM receiving antenatal care from one Baby-Friendly hospital located in Bangkok, Thailand. Since this is a Baby-Friendly Hospital, healthcare clinicians may be more interested in improving and sustaining breastfeeding. Our research team will identify potential participants from the electronic health record (EHR). Recruitment will be achieved through face-to-face outreach in a private room at the clinic or over the phone.

Purposive sampling will be used to enroll participants. Women will be eligible to join if they have experience with T2DM self-management during pregnancy [28]. The inclusion criteria will be 1) pregnant women with preexisting T2DM, defined as either pregestational type 2 or overt diabetes mellitus requiring medical treatment such as oral agents or insulin injection, 2) either primigravida or multigravida, 3) reproductive age (20–44 years old), 4) able to speak the Thai language, and 5) able to provide consent. All participants have been diagnosed with T2DM prior to being enrolled in our study and thus will have had some education and prior diabetes self-management experiences. The exclusion criteria will be pregnant women with any significant complications such as blindness or life-threatening illnesses such as myocardial infarction. After enrollment, if a woman develops diabetes complications such as retinopathy, she will not be dropped from the study unless she wishes to withdraw from our study. If a woman withdraws, we will recruit another participant.

Our goal is to seek a maximally diverse group of pregnant women with T2DM using maximum variation sampling [28] to capture various experiences that emerge from women's different baseline of HbA1c level, gravidity, and trimester. Our above criteria were derived from the literature as we intend to capture a comprehensive description of women who have various

experiences with their diabetes self-management in pregnancy across trimesters and their breastfeeding in postpartum [29, 30].

Pregnant women receive the maternal and child health book or "Pink Book" when they access the antenatal clinic during their first visit. The Pink Book provides health information and health services from pre-conception until the child is 5 years old. Additionally, the clinic services include counseling on breastfeeding and nutrition.

## Ethical considerations

Ethical approval has been obtained from the University of North Carolina at Chapel Hill (IRB: 21–1477) and the Mahidol University in Thailand (IRB number 3428).

Before the potential participants decide to participate in this study, either the principal investigator (PI) or a trained research assistant will explain the objectives, data collection procedures, benefits, and impact of this study. Participants will be informed in the written consent that data from their EHR will be used in the study. Written informed consent will be obtained during participants' enrollment.

## Sample size

As this study is descriptive rather than comparative in nature, statistical significance does not drive the determination of target sample size. The sample size will initially focus on conducting approximately 20 semi-structured interviews [31, 32], but the final sample size will be determined by data saturation [28].

## Instruments

**Interview guide.**   Our research team developed an interview guide in collaboration with experts in women's health and diabetes self-management. The interview guide lists open-ended and follow-up questions focusing on pre-pregnancy care, diabetes self-management, and breastfeeding. The interview questions were first developed in English by the PI (a native Thai, fluent English speaker) and five researchers who are native English speakers and have expertise in maternal health and diabetes management. Then, one bilingual language researcher translated the interview guide from English to Thai. Another native Thai speaking researcher then evaluated the interview guide's clarity and cultural appropriateness. The PI then re-read both Thai and English versions to confirm the interview guide's consistency and accuracy. Before study implementation, the interview guide will be pilot tested with two pregnant women with T2DM in Thailand. The interviewer will ask for feedback on the clarity of interview questions and probes, and the interview guide will be revised as needed. Each woman will be asked the same questions in the same order (S1 File).

**Questionnaires.**   For Time 1, each woman will response to three questionnaires as follows.

A demographic questionnaire includes the mother's age, marital status, educational level, monthly household income, employment status, duration of diabetes, trimester of pregnancy, comorbid conditions, and current medications.

We will use the Thai version of the 14-item Breastfeeding Self-Efficacy Scale-Short Form (BSES-SF; Cronbach's alpha = 0.84) to measure a woman's confidence to breastfeed her infant [33]. The BSES-SF use a 5-point Likert-type scale where 1 = not at all confident and 5 = very confident. Item ratings will be summed to produce a total score from 14 to 70, with higher scores indicating higher confidence. The BSES-SF is as good as the long version and culturally appropriate for Thai women [33]. The rationale for using the short version is to reduce participants' burden.

We will use the Thai version of the 5-item Infant Feeding Intentions Scale (T-IFI; Cronbach's alpha = 0.86) to measure maternal intention to breastfeed her infant [34]. The original Cronbach's alpha was 0.90 [35]. The T-IFI use a 5-point Likert-type scale, individually scored from 0 to 4. Total IFI score is calculated by averaging the score for the first two items and summing this average with the scores for items 3–5. The possible score ranges from 0 to 16, with 0 representing a very strong intention not to breastfeed and 16 representing a very strong intention to fully breastfeed as the sole source of milk until six months of age.

**Physiological measures.** Upon enrollment into the study, a woman's gestation will be established, and biomarker data will be collected retrospectively and prospectively according to the schedule detailed in S2 File. We acquire a woman's physiological measures from the EHR as the objective data which will be used for triangulation with subjective data from the interview.

Biomarker data include maternal body mass index (BMI), gestation weight gain (GWG), HbA1c, and fasting plasma glucose (FPG). Thai women are advised and encouraged to attend the four-visit focused ANC (FANC) model at 8–12 weeks, 24–26 weeks, 32 weeks, and 36–38 weeks of gestation. We will also collect the HbA1c level; a woman's average blood glucose for the past two to three months. Additionally, HbA1c level from prepregnancy and at 24–26 weeks ANC visit will be collected.

We will extract maternal FPG levels measured at 4–6 weeks postpartum from the EHR. FPG is a woman's blood glucose level after fasting at least eight hours before the test. The target level is 80–100 mg/dl [6]. We collect the FPG level because it is recommended for assessing glycemic control in early postpartum and essential for creating a future diabetes care plan [6].

## Data collections

**Time 1 data collection.** At time 1 the pregnant women will complete questionnaires and engage in an interview about diabetes self-management and breastfeeding intention. Data collection will take 1–1½ hours. Participants will first be asked to fill out the demographic, BSES-SF, and T-IFI questionnaires, which will take approximately 15 minutes. The participants will then participate in an interview at that time or later either by phone, Zoom call with no video on/phone access only, or in-person according to their preference. The interview appointment is based on a woman's availability for a 60-minute interview. In-person recorded interviews will be conducted in a private room in the antenatal clinic, and each interview will take approximately 60 minutes. The interviewer will follow an interview guide. The interviewer will write field notes about the environment and participant observation during the interview. Each field note includes interview date and time as well as the mode of interview. We will incorporate our field notes into our qualitative data analysis to articulate any differences according to interview mode and whether the interview conducted immediately after the quantitative part or later.

For virtual interviews, phone or a Zoom call with no video on/phone access only will be used. For phone interviews, the Voice Memos app will be used to audio-record the interview. The Zoom or phone recordings will not include names, telephone numbers, medical record numbers, or other identifiable items. The quality of the recording will be reviewed after each interview. The interviewer will then complete reflective notes to summarize the interview's overall quality and factors contributing to interview quality for every interview regardless of interview means.

**Contacts between time 1 and time 2.** At time 1, we will ask for a woman's and a couple of her family members' phone numbers and permission to call family members if we cannot contact the participant. To maximize retention, we will do monthly check-in calls according to a

woman's preferred communication mode (a phone call or a text message) until the time 2 data collection.

**Time 2 data collection.** Time 2 data collection is at 4–6 weeks postpartum and is an open-ended interview about breastfeeding experiences and infant feeding methods such as exclusive breastfeeding (e.g., direct breastfeeding and expressed breast milk) and non-exclusive breastfeeding (e.g., formula feeding). The second interview will be conducted by phone. Each interview will take approximately 60 minutes. A mother's fasting plasma glucose will be measured at the 4–6 weeks postpartum visit and extracted from the EHR.

Participants will receive a $10 gift card after each data collection section for a total of $20. The schedule of data collection procedures and forms is summarized in Table 1.

## Data management

For qualitative data, we will transcribe the audiotape verbatim in the Thai language. To ensure the transcription quality, the interviewer will review the Thai audio recordings along with the electronic transcript. Additional information (e.g., silences, pauses, and laughing) will be noted in each transcribed interview. All identifying information will be removed, and it will

**Table 1. Schedule of procedures and forms.**

| Procedures/Forms | Data Source | Recruitment | T1 | FANC* | T2 | Alpha |
|---|---|---|---|---|---|---|
| Eligibility screening | PI, RA, RNs | X | | | | |
| Enrollment | PI, RA | | X | | | |
| Inform consent | Participants | | X | | | |
| Demographic | Participants | | X | | | |
| **Aim 1** | | | | | | |
| First interviews | Participants | | X | | | |
| HbA1c (prepregnancy) | Chart review | | X | | | |
| **Aim 2** | | | | | | |
| First interviews | Participants | | X | | | |
| HbA1c (pregnancy) | Chart review | | X | | | |
| Body mass index | Chart review | | X | XXX | | |
| Gestational weight gain | Chart review | | X | XXX | | |
| **Aim 3** | | | | | | |
| First interviews | Participants | | X | | | |
| BSES-SF | Participants | | X | | | .84 |
| T-IFI | Participants | | X | | | .86 |
| **Aim 4** | | | | | | |
| Second interviews | Participants | | | | X | |
| Fasting plasma glucose | Chart review | | | | X | |
| Infant feeding methods | Participants | | | | X | |
| **Rigor and trustworthiness** | | | | | | |
| Field notes | PI | | X | | X | |
| Member checking | Participants | | X | | | |

*Note.* T1 = Pregnancy; FANC = Four-visit focused antenatal care; T2 = One-month postpartum; PI = Principal Investigator; RNs = Registered Nurses; T-IFI = Thai Infant Feeding Intention Scale; BSES-SF = Breastfeeding Self-Efficacy Scale-Short Form; HbA1c = Glycated Hemoglobin

*Timing of FANC may vary based on a woman's visit schedule and enrollment, but body mass index and gestational weight gain must be collected by chart review at least once.

then be saved as a Microsoft Word file. Second, the transcribed interview will be sent to a bilingual professional translation service to translate into English.

By using a Research Electronic Data Capture (REDCap) mobile to collect quantitative data, all data will be automatically entered into a REDCap database. We will track participants using case identification (ID) numbers. We will store the participants' names, telephone numbers, and EHR IDs in a separate file. We will store interview data (audio files) and transcribed interviews on a secure server. We will create an audit trail with a transparent description of our study, with field notes, data collection processes, and data analysis decisions. Assurances of protection of individual participant privacy and confidentiality will be given. Data results will be reported in aggregate form only.

## Data analysis

**Qualitative data.**   Qualitative data analysis will occur concurrently with data collection as an iterative process. First, English transcripts will be analyzed using Atlas.ti version 9, which will facilitate data organization and joint analysis of the text. Data analysis will be based on directed content analysis [36]. The NIMHD framework's domains (sociocultural environment and behavioral influences at individual, interpersonal, and community levels) will be used to guide initial coding. Two coders will code interview transcripts independently. We will identify recurring patterns of participants' words within the data based on relevance to the predetermined domains of inquiry. Data that cannot be coded as one of the NIMHD framework's domains will be identified and analyzed later to determine if they represent a new category or a subcategory of an existing code [36]. Descriptive coding will be used to assign a single code to a chunk of data (2–3 lines) with stand-alone meaning. The coders will analyze the codes and investigate how different codes can be combined to form an overarching theme, with the visual representation using a mind map from Atlas.ti.

The analysis will involve a constant moving back and forth between the entire dataset and individual transcript coding. The coders will work systematically through the dataset, give equal attention to each transcript, and identify interesting aspects that could form the basis of repeated patterns across the dataset. The coders will then describe a clear definition and name for each theme by identifying each theme's essence and determining each theme's aspect that could capture diabetes self-management and breastfeeding experiences. A table will be developed to present themes, subthemes, and quotations.

Throughout data analysis, the coders will write memos, writing down ideas and potential coding schemes. A record of all decisions made in the data analysis process will be recorded in the audit trail [37]. The two coders will have weekly meetings to assess the interview process, review the transcripts, and evaluate amount of new information to confirm when data saturation is achieved. Discrepancies will be critically examined and compared against the original transcripts and field notes until a consensus is reached. Discrepancies will be also solved by a third researcher who has the expertise in qualitative methods.

**Quantitative data.**   Data from the questionnaires and EHR extractions will be entered and analyzed using IBM SPSS version 25.0. Data will be screened for inconsistent or abnormal values such as 0 or 999 values for blood sugar levels. Continuous measures will be assessed for normality and outliers. Analyses include calculation of means, standard deviations, medians, and ranges for continuous variables and frequencies and percentages for categorical variables.

**Triangulation of qualitative and quantitative data sources.**   Triangulation is desirable in mixed methods research because it serves as validation and confirmation of the phenomena being studied [38]. To conduct triangulation, qualitative and quantitative analyses will be

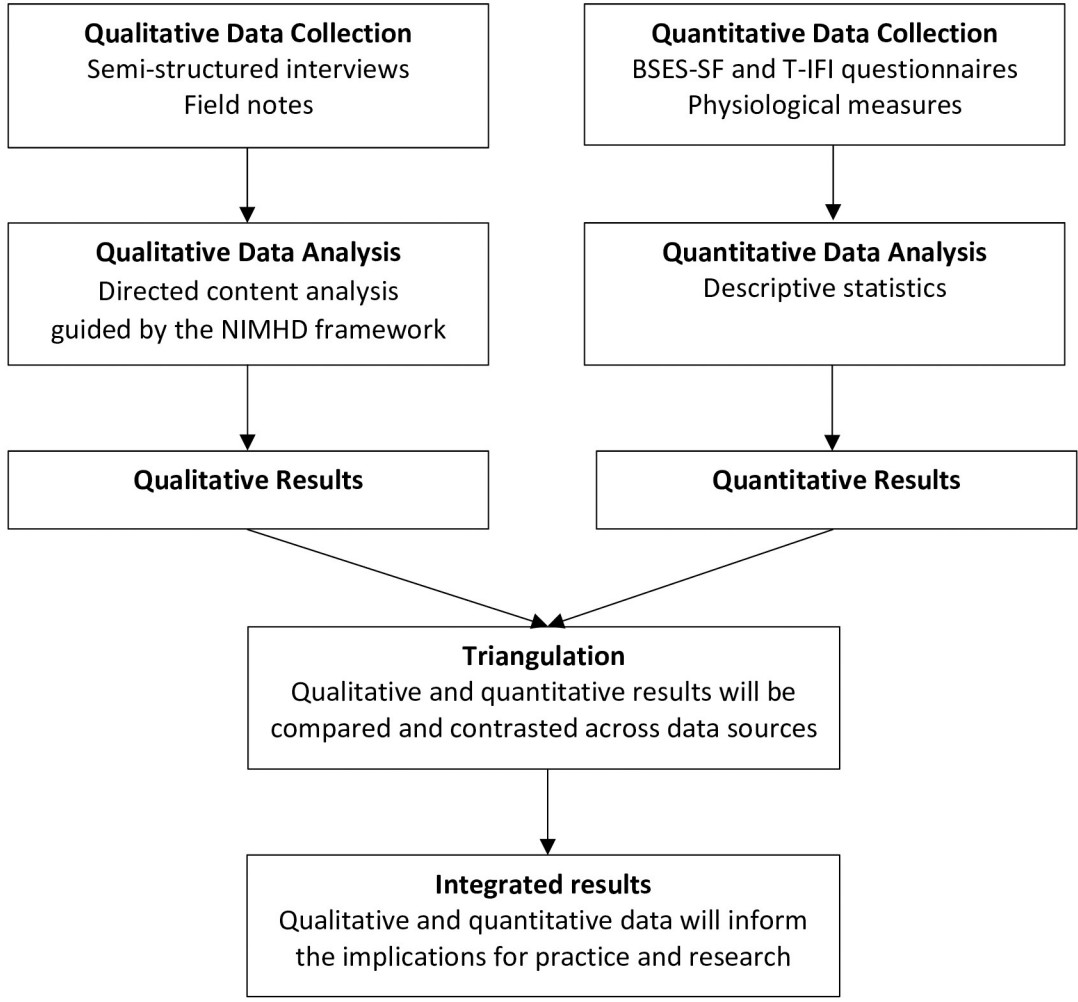

**Fig 1. The triangulation process in the convergent parallel mixed-methods study.**

conducted separately. The results will then be compared and contrasted to identify data convergence and divergence (Fig 1) [24].

The results will be presented in a side-by-side joint display (Table 2). Qualitative themes will be shown in column 1, while column 2 represents the quantitative findings (e.g., frequencies, means, and standard deviations). By using a side-by-side comparison, our research team will analyze and triangulate how the findings represent convergence or divergence between the two data sources [24].

## Discussion

The study described in this protocol will provide a better understanding of Thai women's diabetes self-management during pregnancy and breastfeeding practices in their postpartum period. The strength of using a mixed-method approach is to adequately capture multiple dimensions and increase the depth of understanding of women's experiences from both qualitative and quantitative results.

We recognize that our study protocol has limitations. First, the structure of the interview guide may influence the responses of the interviewee because of predetermined interview

**Table 2. Examples for a joint display.**

| Convergent Parallel Mixed-Methods | | Quantitative results (e.g., frequencies, means, and standard deviations) | | | | | |
|---|---|---|---|---|---|---|---|
| | | HbA1c* | BSES-SF | T-IFI | GWG | FPG | EBF |
| Qualitative themes | Attitude toward diabetes | - or + | | | | | |
| | Confidence toward diabetes | - or + | | | | | |
| | Diabetes self-management barriers | - or + | | | - or + | | |
| | Diabetes self-management facilitators | - or + | | | - or + | | |
| | Breastfeeding confidence | | + | | | | |
| | Breastfeeding intention | | | + | | | |
| | Breastfeeding barriers | | | | | - or + | - |
| | Breastfeeding facilitators | | | | | - or + | + |

*Note*: Convergence noted by plus sign (+). Divergence noted by negative sign (-).

HbA1c = Glycated Hemoglobin; BSES-SF = Breastfeeding Self-Efficacy Scale-Short Form; T-IFI: Thai Infant Feeding Intentions Scale; GWG = Gestational Weight Gain; FPG = Fasting Plasma Glucose; EBF = Exclusive Breastfeeding including direct breastfeeding and expressed breast milk

*Timing of HbA1c levels from prepregnancy and pregnancy may vary based on a woman's visit schedule and data availability.

questions. Therefore, we included open-ended questions that will be pilot tested. Second, the process of extracting maternal health outcomes from the EHR may threaten the overall reliability of findings as the research team may extract data inaccurately. Therefore, we have developed a study protocol and an extraction form to recode EHR data. The research team members will be trained to conduct the EHR review. Third, the study's findings will not be able to be generalized because of the small sample size and a single data collection site. Rather, our goal is transferability since our findings can be applied and transferred to other contexts and samples [37].

Implications of the findings for future nursing research include those future studies in pregnant women with T2DM should use longitudinal designs to illuminate the evolution of pregnancy across the trimesters and postpartum trajectory. Better understanding a woman's particular health care needs can inform the utilization of tools to comprehensively assess breastfeeding intention and confidence that contribute to breastfeeding practices after childbirth. Understanding sociocultural and behavioral factors that influence a woman's experiences can eventually inform the development of patient-centered and culturally tailored care to improve women's health. Our findings may be used as a protocol example when new guidelines are tailored for Asian regions and culturally tailored to different cultural beliefs. The findings will eventually inform policy in delivering a more integrative care model to improve maternal and infant health in Thailand.

## Supporting information

**S1 File. Interview guide.**
(DOCX)

**S2 File. EHR extraction form recording maternal health outcomes.**
(DOCX)

## Acknowledgments

The authors would like to thank Diane C. Berry, PhD, ANP-BC, FAANP, FAAN who made significant contributions to conception and design, was involved in early drafting the manuscript, and provided useful feedback and discussions that helped shape this manuscript.

## Author Contributions

**Conceptualization:** Ratchanok Phonyiam, Marianne Baernholdt, Eric A. Hodges.

**Funding acquisition:** Ratchanok Phonyiam.

**Methodology:** Ratchanok Phonyiam, Marianne Baernholdt, Eric A. Hodges.

**Project administration:** Ratchanok Phonyiam.

**Supervision:** Marianne Baernholdt, Eric A. Hodges.

**Writing – original draft:** Ratchanok Phonyiam, Marianne Baernholdt, Eric A. Hodges.

**Writing – review & editing:** Ratchanok Phonyiam, Marianne Baernholdt, Eric A. Hodges.

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
