## [Decision Letter · Decision Letter 0]

24 Nov 2022

PONE-D-22-16726Self-Management in Pregnancy and Breastfeeding in Women with Type 2

Diabetes Mellitus from Thailand: A Convergent Parallel Mixed-Methods Study ProtocolPLOS ONE

Dear Dr. Phonyiam,

Thank you for submitting your manuscript to PLOS ONE. After careful consideration, we feel that it has merit but does not fully meet PLOS ONE’s publication criteria as it currently stands. Therefore, we invite you to submit a revised version of the manuscript that addresses the points raised during the review process.

We look forward to receiving your revised manuscript.

Kind regards,

Ai Theng Cheong

Academic Editor

PLOS ONE

Journal Requirements:

2. Please amend your list of authors on the manuscript to ensure that each author is linked to an affiliation. Authors’ affiliations should reflect the institution where the work was done (if authors moved subsequently, you can also list the new affiliation stating “current affiliation:….” as necessary).

Reviewers' comments:

Reviewer's Responses to Questions

**Comments to the Author**

1. Does the manuscript provide a valid rationale for the proposed study, with clearly identified and justified research questions?

Reviewer #1: Yes

Reviewer #2: Partly

2. Is the protocol technically sound and planned in a manner that will lead to a meaningful outcome and allow testing the stated hypotheses?

Reviewer #1: Partly

Reviewer #2: Partly

3. Is the methodology feasible and described in sufficient detail to allow the work to be replicable?

Reviewer #1: Yes

Reviewer #2: No

4. Have the authors described where all data underlying the findings will be made available when the study is complete?

Reviewer #1: Yes

Reviewer #2: Yes

5. Is the manuscript presented in an intelligible fashion and written in standard English?

Reviewer #1: Yes

Reviewer #2: No

6. Review Comments to the Author

You may also provide optional suggestions and comments to authors that they might find helpful in planning their study.

Reviewer #1: In general, this study seems feasible to be done. However, there are some comments that is needed to be clarified by the author. Kindly see the attached comments for full review.

Reviewer #2: Title:

The term 'breastfeeding' is rather broad. Suggest to write 'breastfeeding experiences' in parallel with the study objectives.

Abstract:

This study has two Major outcomes/ dependant variables which does not seem to be related, as these outcomes are rather major and encompasses many aspects on its own. Suggest to focus on one major outcome, i.e breastfeeding.

"Data will be collected from 20 pregnant women with preexisting type 2 diabetes mellitus in Thailand who are either primigravida or multigravida, aged 20-44 years old, speak the Thai language, and provide consent."- Provide consent- can be stated in methods, not in abstract.

"Using interviews and questionnaires data will be collected two times. The first time is during pregnancy where data on a woman’s diabetes self-management, breastfeeding confidence, and breastfeeding intention will be collected. The second time is 4-6 weeks postpartum and is about her breastfeeding experiences. Maternal health outcomes"- grammatical error.

"The first time is during pregnancy where data on a woman’s diabetes self-management, breastfeeding confidence, and breastfeeding intention will be collected. The second time is 4-6 weeks postpartum and is about her breastfeeding experiences.It is not clear on how data collection is going to be conducted during these two time-points."- Suggest to briefly elaborate which outcome will be assessed using quantitative and qualitative data respectively.

What ‘health outcomes’ in particular? Suggest to state briefly in abstract.

"This proposed study is significant because the findings will have a potential impact on improving two generations' health across a lifespan. Health care providers will apply the findings to deliver holistic care to women with type 2 diabetes mellitus."-There is lack of strong justification provided by author. Suggest to rephrase to highlight the unique contribution of this study.

Introduction

The introduction should not be presented with detailed statistics as such. Suffice to briefly elaborate the complications with appropriate citations.

"These adverse outcomes can be minimized with proper diabetes self-management during pregnancy."- citation needed.

"An appreciation of the cultural influences on how Thai persons with diabetes view their condition facilitated self-management [7], which in turn, may improve maternal and fetal health."- I don’t quite understand this statement.

"Mothers with T2DM are less likely to establish exclusive breastfeeding at hospital discharge (OR 0.4; 95% CI 0.2, 0.8; p = 0.006), compared to women without T2DM [12]."- Refer to my earlier comment on presentation of detailed statistics in introduction which is unnecessary.

"Further, the number of feedings in the first 24 hours was an independent positive predictor of breastfeeding four months postpartum (OR 1.158; 95% CI 1.021–1.314, p = 0.022), while maternal body mass index was a negative predictor (OR 0.848; 95% CI: 0.779–0.922, p < 0.001) [11]."- This statement seemed to be disjointed. Suggest to add on/ elaborate on other breastfeeding challenges that mothers with T2DM may face (rather than highlighting on the number of feeds and BMI which seemed to be irrelevant.

"The six-month EBF rate in Thailand is still far from optimal. Little is known about how diabetes affects a woman’s confidence and intention to breastfeed her infant and her breastfeeding experiences. Gaining a better understanding of Thai women’s diabetes management experiences during pregnancy and breastfeeding practices postpartum will inform evidence‐based clinical approaches to enhance the health of women with T2DM and their children in Thailand."- This paragraph seems to be disjointed. The earlier statement highlights the rate of EBM inThailand, which followed by ‘little is known about how diabetes affects woman’s confidence…’. Suggest to improve the flow of writing with better connections and relevance between statements.

Material and Methods

The aims of the study should be placed prior to the methods section.

With reference to my prior comment, I highly suggest that the research questions on diabetes self-management and breastfeeding are separated into two different studies.

Study Design

"...because we will discover the who, what, and where of diabetes management and breastfeeding experiences by gaining insights from women with T2DM."- Rephrasing needed (grammatical error).

Descriptive design? I am not sure if this is a common/usual term to be used to describe a quantitative study design. Do you mean to say cross-sectional study design?

Participant eligibility and recruitment

What if the women are diagnosed with T2DM but lack of experienced in T2DM self-management? Will these participants be eligible?

"Participants will be excluded if they have any significant complications such as blindness or life-threatening illnesses such as myocardial infarction."- Suggest to rephrase to more academically appropriate terms.

"These interest criteria were derived from the literature focused on pregnant women who will have various experiences on diabetes self-management in pregnancy [23, 24]."- I don’t quite understand this statement. Suggest to rephrase.

Ethical considerations

"Participants' rights to participate or withdraw from the study without any effect on care will be emphasized. The PI will give the women a chance to ask questions before deciding to participate in this study."- Redundant phrases as this is typically stated in patient information sheet.

"All participants will receive a signed copy of the consent form, which includes the PI’s study office number. They will be encouraged to call with any questions or concerns and the phone numbers for the offices of human subject protection for any questions they may have regarding their rights as study participants."- With reference to my prior comment, this statement typically stated in the participant information sheet/ consent form, hence not necessary to be elaborated in detailed in the manuscript.

Sample Size

"...conducting approximately 20 semi-structured interviews"- Do you have any literatures/ references as to why you chose 20 as your approximate sample size?

Interview Guide

"The interview guide lists open-ended and follow-up questions..."- Need to briefly elaborate the content of the interview guide, what aspects does it cover?

Physiological measures

It is not clear as to the purpose of acquiring these physiological measures as it is not stated as one of the study objectives.

Data Collection

"We will give the women a chance to ask questions before deciding whether to participate."- This is a redundant statement as it is not necessary to state in manuscript.

Time 2 data collection

"A mother’s fasting plasma glucose will be measured at the 4-6 weeks postpartum visit."- As per my previous comment, it is not clear as to the indication of this measurement as it is not included in the study objectives.

Data management

"All data will be kept in a locked file cabinet or computer behind a locked door."- This may not be necessary to be reported in a protocol.

Quantitative data

What do the authors meant by ‘inconsistent and abnormal values’?

Discussion

Will be good to elaborate further on this, on how the findings of this study may have an impact on patient-centred care.

7. PLOS authors have the option to publish the peer review history of their article (what does this mean?). If published, this will include your full peer review and any attached files.

Reviewer #1: No

Reviewer #2: No

---

## [Author Response · Author response to Decision Letter 0]

23 Jan 2023

We appreciate your thoughtful comments and suggestions. We believe the manuscript has been improved through the revision process. Thank you for considering our revised manuscript for publication.

---

## [Decision Letter · Decision Letter 1]

11 Apr 2023

PONE-D-22-16726R1Self-management of type 2 diabetes mellitus in pregnancy and breastfeeding experiences among women in Thailand: Study protocolPLOS ONE

Dear Dr. Phonyiam,

Thank you for submitting your manuscript to PLOS ONE. After careful consideration, we feel that it has merit but does not fully meet PLOS ONE’s publication criteria as it currently stands. Therefore, we invite you to submit a revised version of the manuscript that addresses the points raised during the review process.

We look forward to receiving your revised manuscript.

Kind regards,

Ai Theng Cheong

Academic Editor

PLOS ONE

Journal Requirements:

Reviewers' comments:

Reviewer's Responses to Questions

**Comments to the Author**

1. Does the manuscript provide a valid rationale for the proposed study, with clearly identified and justified research questions?

Reviewer #1: Yes

Reviewer #2: Yes

2. Is the protocol technically sound and planned in a manner that will lead to a meaningful outcome and allow testing the stated hypotheses?

Reviewer #1: Yes

Reviewer #2: Yes

3. Is the methodology feasible and described in sufficient detail to allow the work to be replicable?

Reviewer #1: Yes

Reviewer #2: Yes

4. Have the authors described where all data underlying the findings will be made available when the study is complete?

Reviewer #1: Yes

Reviewer #2: Yes

5. Is the manuscript presented in an intelligible fashion and written in standard English?

Reviewer #1: Yes

Reviewer #2: Yes

6. Review Comments to the Author

You may also provide optional suggestions and comments to authors that they might find helpful in planning their study.

Reviewer #1: The authors have addressed most of concern in the first review. There are still minor things to be improved. For further detail please refer attachment.

Reviewer #2: Overall, the authors have managed to address the feedback/ comments. However, I think the whole manuscript still needs to be proof read for minor grammatical correction. For example. in Line 60 and 62, the author wrote as 'a woman will complete questionnaires', 'a woman will be interviewed', this statements give the impression that only one woman will complete the questionnaire or will be interviewed (instead the authors could modify the statement by stating 'study participants will complete questionnaires', or 'eligible participants will be interviewed..' (suggestion).

Otherwise, the authors managed to revise the manuscript based on the comments and suggestions satisfactorily.

7. PLOS authors have the option to publish the peer review history of their article (what does this mean?). If published, this will include your full peer review and any attached files.

Reviewer #1: No

Reviewer #2: No

---

## [Author Response · Author response to Decision Letter 1]

17 May 2023

We appreciate the opportunity to address your comments and to revise this manuscript. We have indicated all revisions by using text changes, and all page and line numbers refer to locations in the revised manuscript (PONE-D-22-16726.R2). We believe the manuscript has been considerably strengthened with these revisions.

---

## [Editor Report · Decision Letter 2]

22 May 2023

Self-management of type 2 diabetes mellitus in pregnancy and breastfeeding experiences among women in Thailand: Study protocol

PONE-D-22-16726R2

Dear Dr. Phonyiam,

We’re pleased to inform you that your manuscript has been judged scientifically suitable for publication and will be formally accepted for publication once it meets all outstanding technical requirements.

Kind regards,

Ai Theng Cheong

Academic Editor

PLOS ONE
---

## [Editor Report · Acceptance letter]

2 Jun 2023

PONE-D-22-16726R2 

Self-management of type 2 diabetes mellitus in pregnancy and breastfeeding experiences among women in Thailand: Study protocol 

Dear Dr. Phonyiam:

I'm pleased to inform you that your manuscript has been deemed suitable for publication in PLOS ONE. Congratulations! Your manuscript is now with our production department. 

Kind regards, 

on behalf of

Dr. Ai Theng Cheong 

Academic Editor

PLOS ONE